# Ethnic Variations in the Levels of Bone Biomarkers (Osteoprostegerin, Receptor Activator of Nuclear Factor Kappa-Β Ligand and Glycoprotein Non-Metastatic Melanoma Protein B) in People with Type 2 Diabetes

**DOI:** 10.3390/biomedicines12051019

**Published:** 2024-05-06

**Authors:** Preethi Cherian, Irina Al-Khairi, Mohamed Abu-Farha, Tahani Alramah, Ahmed N. Albatineh, Doha Alhomaidah, Fayez Safadi, Hamad Ali, Muhammad Abdul-Ghani, Jaakko Tuomilehto, Heikki A. Koistinen, Fahd Al-Mulla, Jehad Abubaker

**Affiliations:** 1Department of Biochemistry and Molecular Biology, Dasman Diabetes Institute, Dasman 15462, Kuwait; preethi.cherian@dasmaninstitute.org (P.C.); irina.alkhairi@dasmaninstitute.org (I.A.-K.); mohamed.abufarha@dasmaninstitute.org (M.A.-F.); tahani.alramah@dasmaninstitute.org (T.A.); 2Department of Translational Research, Dasman Diabetes Institute, Dasman 15462, Kuwait; abdulghani@uthscsa.edu; 3Faculty of Medicine, Kuwait University, Kuwait City 13110, Kuwait; ahmed.albatineh@dasmaninstitute.org; 4Department of Population Health, Dasman Diabetes Institute, Dasman 15462, Kuwait; doha.alhomaidah@dasmaninstitute.org; 5Department of Anatomy and Neurobiology, Northeast Ohio Medical University, Rootstown, OH 44272, USA; fsafadi@neomed.edu; 6Rebecca D. Considine Research Institute, Akron Children Hospital, Akron, OH 44308, USA; 7Department of Medical Laboratory Sciences, Faculty of Allied Health Sciences, Health Sciences Center, Kuwait University, Kuwait 15462, Kuwait; hamad.ali@ku.edu.kw; 8Division of Diabetes, University of Texas Health Science Center, San Antonio, TX 78030, USA; 9Department of Public Health and Welfare, Finnish Institute for Health and Welfare, 00271 Helsinki, Finland; tuomilehto@hotmail.com (J.T.); heikki.koistinen@helsinki.fi (H.A.K.); 10Saudi Diabetes Research Group, King Abdulaziz University, Jeddah 21589, Saudi Arabia; 11Department of Medicine, University of Helsinki and Helsinki University Hospital, P.O. Box 340, 00029 Helsinki, Finland; 12Minerva Foundation Institute for Medical Research, 00290 Helsinki, Finland

**Keywords:** bone markers, OPG, GPNMB, RANKL, type 2 diabetes, obesity

## Abstract

The global incidence of Type 2 diabetes (T2D) is on the rise, fueled by factors such as obesity, sedentary lifestyles, socio-economic factors, and ethnic backgrounds. T2D is a multifaceted condition often associated with various health complications, including adverse effects on bone health. This study aims to assess key biomarkers linked to bone health and remodeling—Osteoprotegerin (OPG), Receptor Activator of Nuclear Factor Kappa-Β Ligand (RANKL), and Glycoprotein Non-Metastatic Melanoma Protein B (GPNMB)—among individuals with diabetes while exploring the impact of ethnicity on these biomarkers. A cross-sectional analysis was conducted on a cohort of 2083 individuals from diverse ethnic backgrounds residing in Kuwait. The results indicate significantly elevated levels of these markers in individuals with T2D compared to non-diabetic counterparts, with OPG at 826.47 (405.8) pg/mL, RANKL at 9.25 (17.3) pg/mL, and GPNMB at 21.44 (7) ng/mL versus 653.75 (231.7) pg/mL, 0.21 (9.94) pg/mL, and 18.65 (5) ng/mL in non-diabetic individuals, respectively. Notably, this elevation was consistent across Arab and Asian populations, except for lower levels of RANKL observed in Arabs with T2D. Furthermore, a positive and significant correlation between OPG and GPNMB was observed regardless of ethnicity or diabetes status, with the strongest correlation (r = 0.473, *p* < 0.001) found among Arab individuals with T2D. Similarly, a positive and significant correlation between GPNMB and RANKL was noted among Asian individuals with T2D (r = 0.401, *p* = 0.001). Interestingly, a significant inverse correlation was detected between OPG and RANKL in non-diabetic Arab individuals. These findings highlight dysregulation in bone remodeling markers among individuals with T2D and emphasize the importance of considering ethnic variations in T2D-related complications. The performance of further studies is warranted to understand the underlying mechanisms and develop interventions based on ethnicity for personalized treatment approaches.

## 1. Introduction

The gulf region, particularly Kuwait, suffers from high rates of diabetes and obesity. According to a survey conducted in Kuwait in 2014, the prevalence of diabetes was 20% [1]. This high prevalence could be linked to a diet that is high in carbohydrates and simple sugars, accompanied by a sedentary lifestyle leading to a high prevalence of obesity [1]. In addition, family history and genetic background could also contribute to the increased rates of T2D in this region [2,3,4].

Diabetes is a complex disease that impacts several organs of the body since high glucose concentrations in circulating blood reach all parts of the body. There is increasing evidence suggesting a complex interplay between bone metabolism, adipose tissue function, and glucose homeostasis. The detrimental effect of T2D on bone health is now well recognized [5,6,7]. Individuals with T2D were found to have normal to elevated levels of bone mineral density, yet they exhibit up to a threefold higher risk of suffering a fracture [8]. The occurrence of fractures without any changes in bone density could be a result of the dysregulation of markers associated with bone remodeling [9]. It is recommended to incorporate different diagnostic tools to assess bone health in patients with T2D, aiming to understand the intricate interplay of different factors affecting bone health [10]. The bone-remodeling cycle is an intricate process involving the replacement of old and damaged bone through osteoclastic resorption and osteoblastic bone formation. This cycle occurs in sequential steps involving activation, resorption, reversal, formation, and termination [11]. Alternatively, Ma YHV et. al. reported that microvascular issues that affect bone vasculature lead to increased bone marrow adiposity. This may play a vital role in skeletal changes that increase the risk of fracturing and delays in fracture healing seen in T2D [12]. Understanding the cellular and molecular mechanisms that are essential for maintaining bone integrity and mineral homeostasis is pivotal to formulating targets for pharmacological interventions. Our group has been interested in studying the association between diabetes and bone. We have shown that there was an increase in the levels of several bone remodeling markers in individuals with T2D and obesity [13]. Specifically, the highest increase was observed in the circulating levels of Osteoprotegerin (OPG) and Osteoactivin, also known as Glycoprotein non-metastatic melanoma protein B (GPNMB), among people with T2D [13].

OPG, which is a member of the TNF receptor super family, is a decoy receptor for RANKL. Originally, OPG was characterized by its ability to suppress osteoclast formation by preventing the binding of RANKL to its receptor activator of NFĸB (RANK) on the osteoclast precursor cells [14]. During the process of normal bone turnover, the osteoblasts produce a receptor activator of the NFĸB ligand (RANKL), which belongs to the tumor necrosis factor (TNF) family, leading to the induction of osteoclastogenesis [15]. It is well established that the mechanisms underlying osteoporosis involve the OPG-RANKL-RANK axis [16]. Furthermore, recent studies highlight a possible role of the OPG-RANKL-RANK axis in relation to adipose tissue and the regulation of glucose homeostasis [17]. Several studies have identified the roles of OPG, RANKL, and RANK in pancreatic β-cells. It has been reported that OPG stimulated β-cell proliferation by inhibiting RANKL-RANK interaction, thus acting as a mitogen for these cells [18]. It has also been shown that OPG acts as an autocrine or paracrine survival factor that protects β-cells from damage caused by increased cytokine production due to diabetes [19]. Interestingly, a recent study proposed a potential role of the OPG-RANKL-RANK axis in muscle metabolism. This study found that RNKL could induce insulin resistance in muscle cells. On the other hand, inhibiting RANKL with denosumab (Dmab), or an OPG immunoglobulin fragment complex (OPG-Fc) led to enhanced muscle insulin sensitivity and glucose uptake [20]. Furthermore, it was shown that inhibiting RANKL using recombinant mouse OPG (Amgen) or via vector overexpression [21] in a mouse model of T2D led to the improvement of both hepatic insulin resistance and serum glucose concentrations [22,23]. These studies suggest RANKL and OPG act as important regulators of glucose metabolism in pancreas or peripheral tissues [24]. Our previous data demonstrated increased levels of OPG in people with T2D. A similar trend was reported in a study on Chinese post-menopausal women [13,25]. In the current study, the level of RANKL was also investigated in people with T2D, and the relationship between OPG and RANKL was also explored.

The other marker investigated was GPNMB, which was first identified and isolated in melanoma cells [26]. It is a heavily glycosylated type I transmembrane protein expressed in osteoblasts, osteoclasts, macrophages, and dendritic cells [27]. It plays regulatory roles in cell proliferation adhesion and invasion [26]. The interest in GPNMB in relation to bone remodeling is based on its high expression during osteoblast differentiation [28,29]. Our previous data along with other recent studies have demonstrated increased circulating levels of GPNMB in those with metabolic diseases [13,30,31]. A distinct correlation has been established between GPNMB and obesity, wherein GPNMB plays a protective role, with anti-inflammatory effects preventing the progression of obesity-related metabolic complications [32]. GPNMB can be proteolytically cleaved into a soluble molecule by lysosomal stress [33]. The soluble GPNMB is suggested to have an opposite effect, stimulating lipogenesis in white adipose tissue (WAT), resulting in a positive correlation between soluble GPNMB and obesity. This explains the important role of this protein in metabolic disorders [32]. Interestingly, some studies showed that transgenic mice that overexpressed GPNMB showed increased bone resorption [34]. This increased bone resorption leads to bone loss and a weakened bone structure, making bones more prone to fracturing [35]. Hence, the dysregulation of the level of GPNMB seen in obesity and T2D could be a vital factor affecting bone health.

In our current study, we aimed to confirm our previous findings on bone markers in a larger cohort. Additionally, since the population in Kuwait comprises people from various ethnic backgrounds, predominantly Asians and Arabs, we wanted to examine if these changes would apply to individuals belonging to different ethnicities.

## 2. Materials and Methods

### 2.1. Participants and Study Design

The current study is a cross-sectional analysis of data from the Kuwait Diabetes Epidemiological Program (KDEP), which recruited a representative sample of adults (aged ≥ 18) living in Kuwait. This study received approval from the Ethical Review Committee of Dasman Diabetes Institute (DDI) (Protocol number RA2011-003) and was conducted in accordance with the principles of the Declaration of Helsinki. Participants were recruited using a stratified random sampling method designed to ensure a balanced representation across all seven governorates of Kuwait. A list of Kuwaiti residents, complete with their unique identification codes, was provided by the National Public Authority of Civil Information. The selected participants were provided a survey designed using the WHO STEPwise approach to surveillance (STEPS) methodology, as previously reported [36,37]. All participants provided written informed consent before their participation in this study. The primary criteria for exclusion from the study were refusal to sign the consent form, being younger than 18 years of age or older than 65 years of age, or suffering from an ongoing infection. Recruitment took place at DDI, a medical care facility with clinics for diabetes management, a research center, a fitness rehab center, and a radio-imaging facility. It is a well-equipped facility that provides multidisciplinary care and offers advanced therapeutic services. Recruitment took place between April 2011 and June 2014, with a dedicated team consisting of nurses, coordinators, interviewers, and phlebotomists.

### 2.2. Anthropometry and Vital Sign Measurements

Anthropometric measurements, including body weight, height, and waist circumference (WC), were recorded for each participant. Height and weight were measured while the participants were dressed in lightweight indoor clothing and barefoot using calibrated portable electronic weighing scales and portable inflexible height-measuring bars. WC was determined using constant-tension tape at the conclusion of a normal exhalation, with the arms in a relaxed position at the sides; the measurement was taken at the highest point of the iliac crest and at the mid-axillary line. The patients’ body mass index (BMI) values were calculated using the standard formula: body weight (in kilograms) divided by the square of height (in meters).

### 2.3. Laboratory Measurements

Blood samples were collected after confirming that the participants had fasted for at least 10 h overnight. Blood samples were collected in Vacutainer EDTA aprotinin tubes. Plasma was obtained after centrifugation for 10 min at 2000× *g* at room temperature. Subsequently, plasma was aliquoted into cryogenic tubes and stored at −80 °C. Prior to analysis, plasma samples stored in −80 °C freezers were thawed on ice and then centrifuged at 10,000× *g* for 5 min at 4 °C to remove any debris. Blood samples were used to acquire lipid and glycemic profiles, including for fasting plasma glucose (FPG), hemoglobin A1c (HbA1c), fasting insulin, triglycerides (TG), total cholesterol (TC), low-density lipoprotein (LDL), and high-density lipoprotein (HDL). Glucose and lipid profiles were assessed using a Siemens Dimension RXL chemistry analyzer (Diamond Diagnostics, Holliston, MA, USA), whereas HbA1c levels were determined using a VariantTM device (BioRad, Hercules, CA, USA). All laboratory assessments were conducted by certified technicians at the clinical laboratories of DDI, following approved methods and quality standards established by the Ministry of Health. Insulin levels were quantified using the Access Ultrasensitive Insulin Assay (Beckman Coulter, Brea, CA, USA), with both intra- and inter-assay coefficients of variation not exceeding 6%. Insulin resistance was calculated using the Homeostatic Model Assessment for Insulin Resistance (HOMA-IR) formula: (FPG in mmol/L) × (fasting insulin in mU/L)/22.5. Values of HOMA-IR ≤ 2 are classified as normal and HOMA-IR > 2 as insulin resistance [38].

### 2.4. Detection of OPG, RANKL, and GPNMB Plasma Levels Using R&D Custom Multiplexing Assay

Plasma samples were aliquoted into plates and stored at 80 °C until used for the assay. For the multiplexing analysis, plasma samples were thawed and diluted 2X following the kit instructions for the Luminex custom-made panel (cat #LXSAHM, R&D, Los Angeles, CA, USA). The procedure was performed according to the kit’s instructions. In summary, plasma samples were diluted with the sample buffer provided in the kit. Plasma levels of bone markers were assessed using the multiplexing immunobead array. Data were analyzed using Bio-Plex manager software 6.0 (Bio-Rad, Hercules, CA, USA). Results were computed using the 5-PL nonlinear standard curve setting within Bio-Plex Manager software version 6.0. Intra-assay coefficients of variation varied from 1.2% to 3.8%, while inter-assay coefficients of variation ranged from 6.8% to 10.2%. All the above-mentioned assays were carried out according to the instructions of the manufacturers.

### 2.5. Statistical Analysis

Statistical Analysis was performed using SPSS statistical software version 28 (IBM Corp, Armonk, NY, USA). Data were cleaned and checked for any abnormalities. Continuous variables were reported as the median and interquartile range (IQR) to represent the center and variability due to non-normality, while categorical variables were reported as counts and percentages. The non-parametric Mann–Whitney U test was used to compare medians due to the non-normality of the covariates. To compare the medians of a continuous outcome for a factor of three or more levels, the non-parametric Kruskal–Wallis test was used due to either non-normality or the heterogeneity of the variances across groups. To test the association between two categorical variables, the Pearson chi-square test of independence was employed if the expected counts in 80% of the cells were more than 5; otherwise, the Fisher exact test was employed. To measure the strength of the linear relationship between two continuous covariates, correlation analysis was performed with the Spearman’s coefficient to reduce the effect of outliers. All tests were two-tailed and considered statistically significant at *p* ≤ 0.05.

## 3. Results

### 3.1. Characteristics of the Population under Study

A descriptive analysis of our study cohort revealed a predominance of male participants (55.7%), with a median age of 45 years (range: 18–82 years, IQR = 16), including 36% under the age of 40. The ethnic composition was nearly balanced, with 46.6% Arabs and 53.4% Asians. The Arab population primarily consisted of people of Kuwaiti origin (65.4%), while the Asian population predominantly comprised people from India and the Philippines (48.9% and 34.1%, respectively), as shown in Appendix A. This population diversity primarily stems from Kuwait being a gulf country renowned for its job opportunities. In terms of health metrics, 30.76% had Type 2 Diabetes Mellitus (T2DM), and there was a significantly stronger association with diabetes status among individuals of Arab ethnicity compared to their Asian counterparts (*p*-value < 0.001). Arabs showed a significantly higher BMI (31.0, IQR = 7.9) when compared to Asians (26.4 IQR-4.9). Additionally, fasting plasma glucose (FPG), HbA1c, and insulin levels were significantly higher among Arabs compared to Asians, as shown in Table 1. The population was further stratified based on ethnicity, diabetes status, and gender, and these data are provided in Table 2.

### 3.2. Expression of Bone Markers in Circulation

People with diabetes had significantly higher median OPG, RANKL, and GPNMB concentrations than those without T2D (Figure 1). There was a significant increase (*p* ≤ 0.001) in median levels of circulating OPG in those with T2D (826.47 (405.8) pg/mL) versus non-diabetic individuals (653.75 (231.7) pg/mL). Similarly, the median level of circulating RANKL in T2D individuals was 9.25 (17.3) pg/mL, while that in non-diabetic individuals was 0.21 (9.94) pg/mL, and the level of GPNMB in circulation in people with T2D was 21.44 (7) ng/mL, while that in non-diabetic individuals was 18.65 (5) ng/mL.

The data stratified by both ethnicity and diabetes status are presented in Figure 2. This analysis revealed that the median GPNMB level in Arab individuals with T2D (21.42 (8) ng/mL) was significantly higher (*p* < 0.001) than that in non-diabetic Arab individuals (18.21 (5) ng/mL). Among the Asian population, individuals with T2D showed a significantly higher (*p* < 0.001) median level of GPNMB (21.42 (8) ng/mL) compared with that for non-diabetic individuals (18.21 (5) ng/mL). When the population was further stratified by gender, the males showed a significantly higher (*p* < 0.001) median level of GPNMB compared to females in the non-diabetic population from both Arab and Asian backgrounds (Arab males = 19.28 (5) vs. females = 17.23 (5); Asian males = 19.8 (6) vs. females = 17.9 (5)). This significant gender difference was not observed among people with T2D. The median level of OPG in Arab individuals with T2D (881.8 (465.7) pg/mL) was significantly higher (*p* < 0.001) than that in those who were not diabetic (643.0 (243) pg/mL). A similar significant increase (*p* < 0.01) in the median level of OPG was observed among Asians with T2D (758.4 (181.1) pg/mL) compared with that for non-diabetic individuals (644.8 (192.1) pg/mL). There was no significant difference in the median level of OPG when the population was further stratified based on gender.

The median RANKL level did not significantly differ between Arab people with T2D (0.01 (13.43) pg/mL) and those without diabetes (5.13 (12.18) pg/mL), but these medians were significantly higher for Asian individuals with T2D (17.3 (63.9) pg/mL) compared to those for with those without diabetes (0.01 (6) pg/mL, *p* < 0.001).

### 3.3. Correlation Analysis

Spearman’s correlation test was carried out, stratified by ethnicity and diabetes status (Table 3). The correlation between OPG and GPNMB was positive and significant irrespective of ethnicity and diabetes status. This correlation was highest (r = 0.473, *p* < 0.001) among Arab people with T2D. Also, the correlation between GPNMB and RANKL was positive and significant among the Asian individuals with T2D (r = 0.401, *p* = 0.001). A significant inverse correlation was observed between OPG and RANKL among Arab non-diabetic people.

## 4. Discussion

In this study, the population was divided into two groups, Arabs and Asians, and significant differences were observed in the characteristics of both groups. Notably, the Arab population was significantly different from the Asian group in terms of age, obesity, and diabetes status. In the Arab group, more individuals were obese and had T2D. The concentrations of the bone-related markers OPG, RANKL, and GPNMB were assessed in a cohort of individuals with and without T2D living in Kuwait. Since the population in Kuwait largely comprises people originating from Arab and Asian backgrounds, the difference in the expression of these markers based on ethnicity was also investigated. A significant increase in the circulating levels of these markers was observed in people with T2D. The levels of OPG and GPNMB in individuals with T2D were significantly increased both in the Arab and Asian people. The levels of RANKL were found to be lower in the Arabs with T2D than in those without, while among the Asians, there was a significant increase in circulating levels of RANKL in people with T2D. The results of this study agree with previous reports, which have shown that circulating levels of OPG are higher in people with diabetes [39,40]. Such an increase was present in both ethnic groups in our study, indicating that OPG is a relevant marker for diabetes. The role of OPG in bone resorption is well established. It is known that OPG protects bone from excessive resorption and thereby inhibits osteoclastogenesis by preventing the interaction of RANKL with its receptor (RANK) [41]. In the current study, the levels of RANKL were also assessed, and the results showed a significant increase in the circulating level of RANKL in people with T2D. However, differential expression was observed in the RANKL level between Arabs and Asians. Among Asians, an increase in the level of RANKL was seen in people with T2D. But an opposite effect was observed among Arabs. The observed differences in RANKL levels suggest that Asians with T2D might be at a higher risk of osteoporosis due to increased bone resorption. Conversely, Arabs with T2D might have a different risk profile for osteoporosis due to their lower RANKL levels and potentially reduced bone resorption. The RANKL-to-OPG ratio reflects bone health, as it indicates the balance between bone formation and resorption [42]. In this study, the RANKL/OPG ratio was analyzed. Interestingly, the RANKL/OPG ratio among Asians was significantly higher in people with T2D compared to non-diabetic individuals. The significantly higher RANKL-to-OPG ratio observed among Asians with T2D suggests an imbalance favoring bone resorption over formation. This imbalance may contribute to increased bone turnover and potentially lead to a higher risk of osteoporosis and bone fractures in individuals with T2D within the Asian population. On the other hand, the ratio of RANKL/OPG in Arabs displayed an opposite trend, wherein there was a higher ratio among the non-diabetic individuals compared to people with T2D. This indicates that ethnic background has a strong influence on the levels of these markers and their ratio.

The association of the so-called RRO (RANK/RANKL/OPG) triad goes beyond maintaining bone homeostasis. Studies involving individuals with heart failure showed an increase in the levels of RANK, RANKL, and OPG, suggesting their involvement in the development of heart failure [42,43]. Interestingly, several reports have shown an increased prevalence of cardiovascular complications among Asian people with T2D compared with that for people from other ethnic backgrounds [44,45]. There are very few studies showing the cardiovascular outcomes for people with T2D in the Arab population. One study reported that the prevalence of T2D and cardiovascular complications was lower among Arabs compared to that for individuals from other ethnic backgrounds [46]. However, it is noteworthy that the overall level of RANKL is much higher among Arabs than among Asians. This may be a plausible reason behind the high prevalence of CVD among Arabs [47]. Additionally, elevated levels of OPG act as a prognostic tool for assessing the risk of coronary artery calcification and the risk of developing CVD [43]. Furthermore, it has been shown that there is an inverse association between the level of OPG and renal function [48]. Elevated levels of OPG may be an early indicator of chronic kidney disease (CKD) [49]. The RANKL/OPG pathway was also found to play a prominent role in periodontal inflammation associated with diabetes, which, in turn, can affect bone homeostasis and be associated with an increased risk of suffering a fracture [45]. Interestingly, a recent meta-analysis reported that while the serum levels of OPG and RANKL did not directly correlate with osteoporosis, the ratio of these two markers demonstrated a direct association with the condition [50]. Our results show a difference in the OPG/RANKL ratio among Arabs and Asians. This may be used in predicting the probability of developing osteoporotic complications among patients with T2D belonging to different ethnic groups. Furthermore, the dysregulation of the levels of OPG and RANKL seen with T2D may be a factor that determines the impact of this disease in the progression of other complications. Our findings provide insights into how ethnic background influences the varying levels of OPG and RANKL.

The level of GPNMB in circulation was also investigated, and an increase in GPNMB was observed among people with T2D. This increase remained consistent among people from both ethnic groups under study. Levels of GPNMB have been shown to be increased in various metabolic disorders. For instance, serum concentrations of GPNMB were reported to be higher in people with NASH (no-alcoholic steatohepatitis) than in individuals with simple steatosis. Thus, it was suggested that GPNMB may be a promising biomarker and therapeutic target for the development and progression of NAFLD [51]. In a recent study, GPNMB was reported to be a risk factor for obesity and diabetes. Elevated circulating levels of GPNMB, secreted by the liver, induced lipogenesis in WAT-exacerbating-diet-induced obesity and insulin resistance [31]. In an earlier study by our group, an increase in the circulating level of GPNMB was observed in people with obesity and T2D [13]. In addition, increased GPNMB in blood circulation was reported to be a potential biomarker in T2- associated cataracts [30]. It has also been reported that increased levels of GPNMB play an important role in bone metastasis and malignant tumors [52]. Therefore, GPNMB could serve as a crucial therapeutic target for alleviating potential complications associated with T2D.

The differential expression as well as association of OPG, RANKL, and GPNMB seen in this study reinforce the relevance of these markers for T2D. Our research highlights the impact of ethnic background on the expression levels of these markers. This observation may help elucidate the variations reported in serum levels of these markers in the context of obesity and T2D. Furthermore, the results from our study raise the possibility that the markers investigated may contribute to the development of different complications associated with metabolic disorders.

This study is a cross-sectional analysis conducted on a large cohort of participants, enabling the establishment of associations rather than causation. By employing rigorous and robust statistical techniques that mitigate the impact of outliers, this study provides confidence in the obtained results. Nevertheless, one of its limitations includes the utilization of a non-probability convenience-sampling method. While our study accounted for age, gender, diabetes status, and BMI, we did not perform any bone-phenotyping tests to evaluate bone density and strength. This exclusion restricts our ability to fully understand the influence of the changes seen in these markers on metabolic comorbidities related to bone. Future research should include bone-profiling data gathered using a DEXA (Dual X-ray absorptiometry) scan, which could provide a more comprehensive understanding of the roles of these markers in bone-related complications.

## 5. Conclusions

This study highlights the differences in the expression levels of OPG, RANKL, and GPNMB in circulation with respect to ethnicity. It also confirms that the levels of these markers are impacted by diabetes status. The differential expression of bone marker levels between Asians and Arabs with T2D highlights the importance of considering ethnicity-specific factors in understanding the pathophysiology and management of T2D-related bone complications. It underscores the need for tailored approaches to treatment and further research to elucidate the underlying mechanisms driving these differences.

## Figures and Tables

**Figure 1 biomedicines-12-01019-f001:**
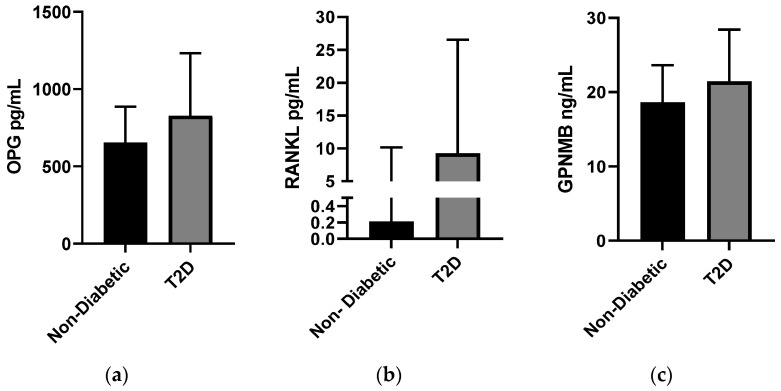
Plasma concentrations of different bone markers in the whole study population (N = 2083) stratified into non-diabetic (n = 1425) and T2D-afflicted (n = 633) individuals. (**a**) Osteoprotegerin (OPG), (**b**) Receptor Activator of Nuclear factor Kappa beta Ligand (RANKL), and (**c**) Glycoprotein non-metastatic b (GPNMB). The levels of bone markers were determined using a multiplex bone panel. The data are presented as a bar graph showing the median values and inter-quartile range. Mann–Whitney U test.

**Figure 2 biomedicines-12-01019-f002:**
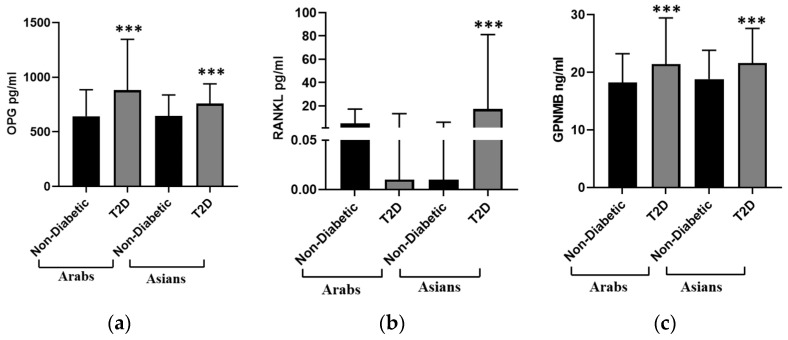
Plasma concentrations of different bone markers stratified according to ethnicity as Arabs (N = 886) and Asians (N = 1020) and comparison of non-diabetic people with people with T2D. (**a**) Osteoprotegerin (OPG), (**b**) Receptor Activator of Nuclear factor Kappa beta Ligand (RANKL), and (**c**) Glycoprotein non-metastatic b (GPNMB). The concentrations of bone markers were determined using a multiplex bone panel. The data are presented as a bar graph showing the median values and inter-quartile range. Mann–Whitney U test; *** *p* <0.001 vs. nondiabetic within respective ethnic group.

**Table 1 biomedicines-12-01019-t001:** Clinical characteristics stratified by ethnicity (N = 2083).

Clinical Characteristic	Arab 46.6%	Asian 53.4%	*p*-Value
Diabetes Status Non-Diabetic (69.24%) T2D (30.76%)	521 (40.9) 365 (57.8)	753 (59.1) 267 (42.2)	<0.001 ^a^
Age, median (IQR)	48 (15)	41 (15)	<0.001 ^b^
Gender, n (%) Male (55.7%) Female (44.3%)	438 (42.0) 461 (52.1)	606 (52.0) 424 (47.9)	<0.001 ^a^
BMI, median (IQR)	31.0 (7.9)	26.4 (4.9)	<0.001 ^b^
Waist-hip ratio, median (IQR)	0.9 (0.08)	0.09 (0.07)	<0.001 ^b^
TC (mmol/L), median (IQR)	5.0 (1.4)	5.2 (1.3)	<0.001 ^b^
HDL (mmol/L), median (IQR)	1.12 (0.4)	1.12 (0.4)	0.317 ^b^
LDL (mmol/L), median (IQR)	3.2 (1.3)	3.4 (1.2)	<0.001 ^b^
TG (mmol/L), median (IQR)	1.3 (0.9)	1.3 (1.0)	0.277 ^b^
FPG (mmol/L), median (IQR)	5.5 (2.2)	5.2 (1.1)	<0.001 ^b^
HbA1c (%), median (IQR)	6.3 (2.4)	6.3 (1.9)	0.023 ^b^
HOMA-IR, median (IQR)	2.0 (2.3)	1.9 (2.0)	0.383 ^b^
Insulin (mU/L), median (IQR)	8.1 (6.9)	7.7 (6.5)	0.005 ^b^

^a^ Point biserial correlation; ^b^ Mann–Whitney U-test, with the *p*-value in parentheses.

**Table 2 biomedicines-12-01019-t002:** Comparison of the levels of the markers of interest and clinical variables stratified by ethnicity and diabetes status (N = 2083).

Variable ^a^	Arabs			Asians		
	Non-Diabetic	Diabetic	*p*-Value ^b^	Non-Diabetic	Diabetic	*p*-Value ^b^
Gender, n (%)			0.058			<0.001
Male	239 (55.6)	191 (44.4)		309 (64.9)	167 (35.1)	
Female	282 (61.8)	174 (38.2)		144 (78.7)	39 (21.3)	
Age, (years)	43.0 (13.0)	55.0 (13.0)	<0.001	39.0 (13)	50.0 (11.5)	<0.001
BMI (Kg)	29.9 (6.9)	32.4 (8.2)	<0.001	26.1 (4.65)	27.4 (5.55)	<0.001
TC (mmol/L)	5.1 (1.4)	4.8 (1.39)	<0.001	5.2 (1.25)	5.2 (1.31)	0.734
HDL (mmol/L)	1.14 (0.46)	1.09 (0.40)	0.003	1.08 (0.40)	1.01 (0.30)	<0.001
LDL (mmol/L)	3.32 (1.10)	2.90 (1.26)	<0.001	3.4 (1.30)	3.30 (1.36)	0.210
TG(mmol/L)	1.20 (0.82)	1.48 (0.98)	<0.001	1.22 (0.87)	1.59 (1.08)	<0.001
FPG (mmol/L)	5.0 (0.70)	7.6 (3.38)	<0.001	4.9 (0.60)	7.60 (3.72)	<0.001
HbA1c (%)	5.5 (0.64)	7.5 (2.35)	<0.001	5.60 (0.70)	7.5 (2.63)	<0.001
HOMA-IR	1.60 (1.31)	3.86 (4.82)	<0.001	1.64 (1.49)	3.93 (3.61)	<0.001
Insulin (mU/L)	7.1 (5.4)	10.7 (10.8)	<0.001	7.49 (6.4)	10.9 (9.32)	<0.001
GPNMB (ng/mL)	18.21 (5)	21.42 (8)	<0.001	18.81 (5)	21.61 (6)	<0.001
OPG (pg/mL)	643.0 (243.3)	881.8 (465.7)	<0.001	644.8 (192.1)	758.4 (181.1)	<0.001
RANKL (pg/mL)	5.13 (12.18)	0.01 (13.43)	0.306	0.01 (6.0)	17.3 (63.9)	<0.001
RANKL/OPG	0.0066 (0.02)	0.0001 (0.01)	<0.001	0.00001 (0.01)	0.0245 (0.04)	<0.001

^a^ Reported as median (IQR) and *p*-values obtained using the Mann–Whitney U test due to non-normality in at least one of the groups; ^b^
*p*-value calculated using Pearson Chi-square test of independence.

**Table 3 biomedicines-12-01019-t003:** Spearman correlations between OPG, GPNMB, and RANKL stratified by ethnicity and diabetes status.

Correlation *	Arab		Asians	
	T2D	Non-Diabetic	T2D	Non-Diabetic
(OPG, GPNMB)	0.473 (<0.001)	0.273 (<0.001)	0.314 (0.010)	0.282 (<0.001)
(OPG, RANKL)	0.020 (0.782)	−0.115 (0.010)	−0.077 (0.543)	0.020 (0.589)
(GPNMB, RANKL)	0.047 (0.516)	0.121 (0.006)	0.401 (0.001)	0.055 (0.133)

* Spearman correlation was used due to many outliers and *p*-values in parentheses.

## Data Availability

The datasets used and/or analyzed during this study are available from the corresponding author on reasonable request.

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
