# Peer review of "Ethnic Variations in the Levels of Bone Biomarkers (Osteoprostegerin, Receptor Activator of Nuclear Factor Kappa-Β Ligand and Glycoprotein Non-Metastatic Melanoma Protein B) in People with Type 2 Diabetes"

_biomedicines, 2024, doi:10.3390/biomedicines12051019_

Round 1

Reviewer 1 Report

Comments and Suggestions for Authors

This manuscript Ethnic Variations in the level of Bone Biomarkers OPG, RANKL and GPNMB in People with Type 2 Diabetes” was aimed to evaluate key biomarkers associated with bone health and remodelling - Osteoprotegerin (OPG), Receptor Activator of Nuclear Factor Kappa-Β Ligand (RANKL), and Glycoprotein Non-Metastatic Melanoma Protein B (GPNMB) - in diabetic individuals, while also examining the influence of ethnicity on these biomarkers. A cross-sectional cohort of 2083 individuals from a multi-ethnic background living in Kuwait was investigated. Since the population in Kuwait is largely comprised of people originating from Arab and Asian backgrounds, the difference in the expression of these markers based on ethnicity was also investigated. The authors concluded that there were the differences in bone remodelling in people with T2D and underscores the importance of considering ethnic variations in T2D-related biomarkers, contributing to a more nuanced understanding of the disease's impact on bone health. Interesting study, large sample, analysing population with type 2 diabetes in region with high and increasing prevalence of this disease, and highlighting the importance of individualisation in type 2 diabetes.

Comments to the Authors:

1.                  Line 154, please add information about medical care level where recruitment took place…  

2.                  Table 2. Please, add units for Variables

3.                  Please add limitation of the study

4.                  Conclusion: please put the main result from the study into conclusions, and then tis reflection about it

Author Response

1. Line 154, please add information about medical care level where recruitment took place… 

Thank you for your suggestion, we have added the details about the facility where the study was conducted. (Lines 158-161, page 4) it reads as follow:

“Recruitment took place at DDI, a medical care facility with clinics for diabetes management, a research center, a fitness rehab center, and a radio imaging facility. It is a well-equipped facility that provides multidisciplinary care and offers advanced therapeutic services.”

2. Table 2. Please, add units for Variables.

    Thank you for your observation. We have added units to the variables in both tables.

3. Please add limitation of the study

Thank you for your kind remark. We have added a paragraph on the limitations of this study (Line 385-395, page 10) and it reads as follows:

“This study is a cross sectional analysis conducted on a large cohort of participants enabling the establishment of associations rather than causation. By employing rigorous and robust statistical techniques that mitigate the impact of outliers, this study provides confidence in the obtained results. Nevertheless, one of the limitations includes the utilization of a non-probability convenience sampling method. While our study accounted for age, gender, diabetes status, and BMI we did not perform any bone phenotyping test to check for bone density and strength. This exclusion restricts our ability to fully understand the influence of the changes seen in these markers with metabolic comorbidities related to bone. Future research should include the bone profiling data using DEXA (Dual x-ray absorptiometry) scan that could provide a more comprehensive understanding on the role of these markers in bone related complications.”

4. Conclusion: please put the main result from the study into conclusions, and then tis reflection about it

    Thank you for your comment. We have modified the conclusion as requested. (Lines 397-403, page 10) and it reads as follows:

“This study highlights the differences in the expression levels of OPG, RANKL and GPNMB in circulation with ethnicity. It also confirms that the levels of these markers are impacted by diabetes status. The differential expression of bone markers levels between Asians and Arabs with T2DM highlights the importance of considering ethnic-specific factors in understanding the pathophysiology and management of T2DM-related bone complications. It underscores the need for tailored approaches to treatment and further research to elucidate the underlying mechanisms driving these differences.”

Reviewer 2 Report

Comments and Suggestions for Authors

1. Incorporate Detailed Methodology: Provide a brief overview of the study's methodology, including details on participant selection criteria, data collection methods, and statistical analyses used. This will help readers better understand the study's approach and strengthen the credibility of the findings.

2. Clarify Ethnic Backgrounds: Specify the ethnic groups represented in the study and any relevant details about their demographics or characteristics. Additionally, discuss if there were any specific reasons for including these ethnicities and if recruitment strategies varied among them.

3. Discuss Clinical Implications: Briefly discuss the clinical implications of the study's findings, particularly regarding the impact of biomarker differences on the management and treatment of diabetes-related bone complications. Consider how these findings could inform clinical practice and personalized treatment approaches.

4. Highlight Limitations: Acknowledge any limitations of the study, such as potential biases, confounding factors, or limitations related to sample size or study design. Addressing these limitations upfront will enhance the transparency and credibility of the study.

5. Propose Future Directions: Conclude the abstract with suggestions for future research directions, such as exploring underlying mechanisms, potential interventions, or personalized medicine approaches based on ethnicity. This will demonstrate the broader significance and implications of the study's findings.

6. Ensure Clarity and Conciseness: Review the abstract to ensure that the language is clear, concise, and accessible to a broad audience. Avoid using unnecessary jargon or technical terms that may hinder understanding.

.

Comments on the Quality of English Language

Some minor revision needs to be made

Author Response

Incorporate Detailed Methodology: Provide a brief overview of the study's methodology, including details on participant selection criteria, data collection methods, and statistical analyses used. This will help readers better understand the study's approach and strengthen the credibility of the findings.

In section 2 – The materials and methods were modified as suggested. Kindly refer to sections 2.1- 2.5 (Lines145-220, page 3-5) 

Clarify Ethnic Backgrounds: Specify the ethnic groups represented in the study and any relevant details about their demographics or characteristics. Additionally, discuss if there were any specific reasons for including these ethnicities and if recruitment strategies varied among them.

Thank you for your interest in knowing the details of our study population. This study included individuals from Arab and Asian background. The details about their demographics are included in Supplementary table 1(S1), the details on their characteristics have been modified in the results section 3.1 (Lines 223-236, page 5) and in table 1 in the manuscript, As mentioned in the discussion section 4 (Lines 299-302, page 8) the study was performed on a cohort of individuals living in Kuwait which is predominantly comprised of people of Arab and Asian origin. Also, the methods section the criteria for selecting participants has been provided (Lines 145-153, page 3)

Discuss Clinical Implications: Briefly discuss the clinical implications of the study’s findings, particularly regarding the impact of biomarker differences on the management and treatment of diabetes-related bone complications. Consider how these findings could inform clinical practice and personalized treatment approaches.

Thank you for your comment which is helpful in improving the quality of our manuscript. We have included a brief discussion on the impact of these markers in diabetes related bone complications (Lines 328-332; 353-358; 374-377; 379-382) as shown below.

“The significantly higher RANKL to OPG ratio observed among Asians with T2DM suggests an imbalance favoring bone resorption over formation. This imbalance may contribute to increased bone turnover and potentially lead to a higher risk of osteoporosis and bone fractures in individuals with T2DM within the Asian population.”

“Interestingly, a recent meta-analysis reported that while the serum levels of OPG and RANKL did not directly correlate with osteoporosis, the ratio of these two markers demonstrated a direct association with the condition [49]. Our results, show a difference in the OPG/RANKL ratio among Arabs and Asians. This may be used in predicting the possibility of developing osteoporotic complications in patients with T2D belonging to different ethnic groups.”

“It has also been reported that increased level of GPNMB plays an important role in bone metastasis and malignant tumors [51]. Therefore, GPNMB could serve as a crucial therapeutic target for alleviating potential complications associated with T2D.”

“Our research highlights the impact of ethnic background on the expression levels of these markers. This observation may help elucidate the variations reported in serum levels of these markers in the context of obesity and T2D.”

We have also revised our conclusion accordingly and it reads as follows:

“This study highlights the differences in the expression levels of OPG, RANKL and GPNMB in circulation with ethnicity. It also confirms that the levels of these markers are impacted by diabetes status. The differential expression of bone markers levels between Asians and Arabs with T2DM highlights the importance of considering ethnic-specific factors in understanding the pathophysiology and management of T2DM-related bone complications. It underscores the need for tailored approaches to treatment and further research to elucidate the underlying mechanisms driving these differences.”

Highlight Limitations: Acknowledge any limitations of the study, such as potential biases, confounding factors, or limitations related to sample size or study design. Addressing these limitations upfront will enhance the transparency and credibility of the study.

Thank you for your kind remark. We have added a paragraph on the limitations of this study (Line 385-395, page 10) and it reads as follows:

“This study is a cross sectional analysis conducted on a large cohort of participants enabling the establishment of associations rather than causation. By employing rigorous and robust statistical techniques that mitigate the impact of outliers, this study provides confidence in the obtained results. Nevertheless, one of the limitations includes the utilization of a non-probability convenience sampling method. While our study accounted for age, gender, diabetes status, and BMI we did not perform any bone phenotyping test to check for bone density and strength. This exclusion restricts our ability to fully understand the influence of the changes seen in these markers with metabolic comorbidities related to bone. Future research should include the bone profiling data using DEXA (Dual x-ray absorptiometry) scan that could provide a more comprehensive understanding on the role of these markers in bone related complications.”

Propose Future Directions: Conclude the abstract with suggestions for future research directions, such as exploring underlying mechanisms, potential interventions, or personalized medicine approaches based on ethnicity. This will demonstrate the broader significance and implications of the study’s findings.

Thank you for your input, we have added this sentence in the abstract (Lines 59-62) and it reads as follows:

“These findings highlight dysregulation in bone remodeling markers among individuals with T2D and emphasize the importance of considering ethnic variations in T2D-related complications. Further studies are warranted to understand the underlying mechanisms and develop interventions based on ethnicity for personalized treatment approaches.”

Ensure Clarity and Conciseness: Review the abstract to ensure that the language is clear, concise, and accessible to a broad audience. Avoid using unnecessary jargon or technical terms that may hinder understanding.

Thank you for your valuable suggestion. We have modified the abstract to be clearer and more concise. (Lines 42-62)

Reviewer 3 Report

Comments and Suggestions for Authors

Authors should:

1. Extend the comments on the differences with the published data

2. Please comment the "contradictory" observation that an increase in bone marrow density coincides with 3-fold risk of fracture

3. Table 1 should be rewritten and split the base lien data of the two populations studied (arabs and asians). Comment of the origin of the asians: are they immmigrants? age? gender? etc

4. Discussion: discuss the differences between the two populations. 

5. Conclusion: change- is there any conclusion on tje ethnicity?

Author Response

Extend the comments on the differences with the published data.

Thank you for your suggestion. We have added the following paragraphs to the result and discussion sections to discuss the ethnic differences:

“However, a differential expression was observed in RANKL level between Arabs and Asians. Among Asians an increase in the level of RANKL was seen in people with T2D. But an opposite effect was observed among Arabs. The observed differences in RANKL levels suggest that Asians with T2DM might be at a higher risk of osteoporosis due to in-creased bone resorption. Conversely, Arabs with T2DM might have a different risk profile for osteoporosis due to lower RANKL levels and potentially reduced bone resorption.” (lines 321-325, page 8)

“In this study the RANK/OPG ratio was analyzed. Interestingly, the RANKL/OPG ratio among Asians was significantly higher in people with T2D as compared to the non-diabetic individuals. The significantly higher RANKL to OPG ratio observed among Asians with T2DM suggests an imbalance favoring bone resorption over formation. This imbalance may contribute to increased bone turnover and potentially lead to a higher risk of osteoporosis and bone fractures in individuals with T2DM within the Asian population.” (Lines 326-333 of pages 8-9)

“Interestingly, a recent meta-analysis reported that while the serum levels of OPG and RANKL did not directly correlate with osteoporosis, the ratio of these two markers demonstrated a direct association with the condition [49]. Our results, show a difference in the OPG/RANKL ratio among Arabs and Asians. This may be used in predicting the possibility of developing osteoporotic complications in patients with T2D belonging to different ethnic groups.” (Lines 353-358 of page 9)

“The differential expression as well as association of OPG, RANKL and GPNMB seen in this study reinforces the relevance of these markers in T2D. Our research highlights the impact of ethnic background on the expression levels of these markers. This observation may help elucidate the variations reported in serum levels of these markers in the context of obesity and T2D. Furthermore, the results from our study raise a possibility that the markers investigated may contribute to the development of different complications associated with metabolic disorders.” (Lines 378-384 of pages 10-11)

Please comment the "contradictory" observation that an increase in bone marrow density coincides with 3-fold risk of fracture.

Ans: Thank you for your recommendation. We have added a comment and reference on the above statement (Lines 79-81, page 2) and it reads as follows:

“The occurrence of fractures without any changes in bone density could be a result of the dysregulation in markers associated with bone remodeling [9]. It is recommended to incorporate different diagnostic tools to assess bone health in patients with T2D aiming to understand the intricate interplay of different factors affecting bone health [10].”

Table 1 should be rewritten and split the base lien data of the two populations studied (arabs and asians). Comment of the origin of the asians: are they immmigrants? age? gender? Etc

Ans: Thank you for your feedback to improve the data presented in our manuscript. We have modified table 1 as recommended. The demographic distribution of the population stratified by ethnicity was also detailed in Table S1. The Asian population predominantly comprised of people from India and the Philippines (48.9% and 34.1% respectively).

3. Discussion: discuss the differences between the two populations. 

We appreciate the reviewer’s comment. The following paragraphs were added to the result and discussion sections detailing the ethnic differences:

Result:

“In terms of health metrics, 30.76% had Type 2 Diabetes Mellitus (T2DM), with a significantly stronger association with diabetes status among individuals of Arab ethnicity compared to their Asian counterparts (p-value < 0.001). Arabs showed a significantly higher BMI (31.0, IQR=7.9) when compared to Asians (26.4 IQR-4.9). Additionally, fasting plasma glucose (FPG) level, HbA1c, and insulin levels were significantly higher among Arabs compared to Asians as shown in Table 1.” (lines 230-235, page 5)

Discussion:

“However, a differential expression was observed in RANKL level between Arabs and Asians. Among Asians an increase in the level of RANKL was seen in people with T2D. But an opposite effect was observed among Arabs. The observed differences in RANKL levels suggest that Asians with T2DM might be at a higher risk of osteoporosis due to in-creased bone resorption. Conversely, Arabs with T2DM might have a different risk profile for osteoporosis due to lower RANKL levels and potentially reduced bone resorption.” (lines 321-325, page 8)

“In this study the RANK/OPG ratio was analyzed. Interestingly, the RANKL/OPG ratio among Asians was significantly higher in people with T2D as compared to the non-diabetic individuals. The significantly higher RANKL to OPG ratio observed among Asians with T2DM suggests an imbalance favoring bone resorption over formation. This imbalance may contribute to increased bone turnover and potentially lead to a higher risk of osteoporosis and bone fractures in individuals with T2DM within the Asian population.” (Lines 326-333 of pages 8-9)

“Interestingly, a recent meta-analysis reported that while the serum levels of OPG and RANKL did not directly correlate with osteoporosis, the ratio of these two markers demonstrated a direct association with the condition [49]. Our results, show a difference in the OPG/RANKL ratio among Arabs and Asians. This may be used in predicting the possibility of developing osteoporotic complications in patients with T2D belonging to different ethnic groups.” (Lines 353-358 of page 9)

“The differential expression as well as association of OPG, RANKL and GPNMB seen in this study reinforces the relevance of these markers in T2D. Our research highlights the impact of ethnic background on the expression levels of these markers. This observation may help elucidate the variations reported in serum levels of these markers in the context of obesity and T2D. Furthermore, the results from our study raise a possibility that the markers investigated may contribute to the development of different complications associated with metabolic disorders.” (Lines 378-384 of pages 10-11)

5. Conclusion: change- is there any conclusion on the ethnicity?

 As advised by the reviewer we have edited the conclusion, and it reads as  follows:

“This study highlights the differences in the expression levels of OPG, RANKL and GPNMB in circulation with ethnicity. It also confirms that the levels of these markers are impacted by diabetes status. The differential expression of bone markers levels between Asians and Arabs with T2DM highlights the importance of considering ethnic-specific factors in understanding the pathophysiology and management of T2DM-related bone complications. It underscores the need for tailored approaches to treatment and further research to elucidate the underlying mechanisms driving these differences.”

Round 2

Reviewer 3 Report

Comments and Suggestions for Authors

This manuscript shows original data, well recorded and may be useful for clinicians who treat type 2 diabetes mellitus and bone remodelling complications.

Author Response

Thank you for your feedback. We greatly appreciate that you find that our manuscript could be of added value to clinicians.